# Monetary Policy Rule and Taylor Principle in Mongolia: GMM and DSGE Approaches

**Hiroyuki Taguchi * and Ganbayar Gunbileg**

Graduate School of Humanities and Social Sciences, Saitama University, 225 Shimo-Okubo, Sakura-ku, Saitama 338-8570, Japan; ganbayar.g.035@ms.saitama-u.ac.jp

* Correspondence: htaguchi@mail.saitama-u.ac.jp; Tel.: +81-48-858-3324

**Abstract:** This article aims to examine the monetary policy rule under an inflation targeting in Mongolia with a focus on its conformity to the Taylor principle, through two kinds of approaches: a monetary policy reaction function by the generalized-method-of-moments (GMM) estimation and a New Keynesian dynamic stochastic general equilibrium (DSGE) model with a small open economy version by the Bayesian estimation. The main findings are summarized as follows. First, the GMM estimation identified an inflation-responsive rule fulfilling the Taylor principle in the recent phase of the Mongolian inflation targeting. Second, the DSGE-model estimation endorsed the GMM estimation by producing a consistent outcome on the Mongolian monetary policy rule. Third, the Mongolian rule was estimated to have a weaker response to inflation than the rules of the other emerging Asian adopters of an inflation targeting.

**Keywords:** monetary policy rule; Taylor principle; Mongolia; inflation targeting; GMM; the New Keynesian DSGE model; E52; E58; O53

## 1. Introduction

Mongolia has evolved her monetary policy framework, since she transformed the economic system from a centrally planned economy to a market-based economy in the early 1990s. In 1991, the Bank of Mongolia (BOM) started implementing monetary policy as a central bank. At the first stage until 2006, the BOM adopted a monetary aggregate targeting with its reserve money being an operational target. Since the mid-2000s, however, the linkage between reserve money and inflation became unstable due to financial deepening processes, and thus the monetary aggregate target lost its effectiveness. Under this background, the BOM introduced an inflation targeting framework in 2007 as the second stage. In this framework, the BOM equipped the policy mandates of announcing a mid-term targeted inflation rate to the public and of taking every possible measures to maintain the inflation rate within its targeted range. At the same time, in July 2007 the BOM adopted the one-week central bank bills' rate as a policy rate, so that the policy rate could work as an operating target to attain its targeted inflation rate. Having received and completed the IMF Stand-by program during the wave of the world financial crisis in 2009, the BOM has taken several steps to upgrade the inflation targeting system as a recent stage. The BOM developed the Forecasting and Policy Analysis System (FPAS) since 2011, aiming at forecast-based policy formation and decision making, and effective communication with the public under the inflation targeting. The BOM has also improved its operational framework by establishing an interest rate corridor to enhance the policy rate transmission mechanism from 2013.

When it comes to an analytical issue on rule-based monetary policies, the "monetary policy reaction function" has been a useful instrument to evaluate monetary policy rules practiced by central banks in a quantitative way. The function, a more generalized form of the so-called Taylor rule proposed by Taylor (1993), describes the policy rules in such a way that central banks adjust their policy

rates in response to the gaps between expected inflation and output and their respective targets. The functions were initially estimated by the seminal work by Clarida et al. (1998) for examining monetary policies of two sets of countries: the G3 (Germany, Japan, and the US) and the E3 (UK, France, and Italy). Since then, the functions have been widely applied for analyzing or describing the monetary policy rules not only in advanced economies but also in emerging-market economies.

In examining the monetary policy reaction function, one of the most crucial criteria to judge the workability of monetary policy rules to control inflation would be whether the rules fulfill the "Taylor principle": for inflation to be stable, the central bank must respond to an increase in inflation with an even greater increase in a nominal interest rate (Mankiw 2016). In case a policy rate's upward reaction to a hike of inflation rate is less than unity, the suppressed "real" interest rate could further accommodate inflation, thereby leading to a vicious circle of spiraling inflation. The Taylor principle is in general considered to hold in advanced economies through Clarida et al. (1998) and the subsequent empirical studies (e.g., Belke and Polleit 2007). For developing and emerging-market economies like Mongolia, however, the validity of the Taylor principle is questionable and has also been limitedly studied, even though the economies adopted an inflation targeting in their monetary policy frameworks.

Another point worth noting in analyzing monetary policy rules in developing and emerging-market economies is that their rules are often supposed to take into account not only inflation and output gap but also exchange rate fluctuations. Calvo and Reinhart (2002) argued that there seemed to be an epidemic case of the "fear of floating", particularly among emerging-market economies. The fear of floating comes from a lack of confidence in currency value, especially given that their external debt is primarily denominated in US dollars, which is often referred to as the "original sin" hypothesis (Eichengreen and Hausmann 1999). The principle of the "impossible trinity", on the other hand, demonstrates that an economy has to give up one of three goals: fixed exchange rate, independent monetary policy, and free capital flows. Thus, given the capital mobility, emerging-market economies tend to face the trade-off in their policy targets between exchange-rate stability and price stability: their efforts to avoid exchange rate volatility prevent their monetary authorities from concentrating fully on an inflation targeting.

This article aims to examine the monetary policy rule under an inflation targeting in Mongolia with a focus on its conformity to the Taylor principle, through the two kinds of models: a monetary policy reaction function by the generalized-method-of-moments (GMM) estimation and a New Keynesian dynamic stochastic general equilibrium (DSGE) model by the Bayesian estimation. The contributions of this study are summarized as follows. First, this study focuses on the conformity to the Taylor principle in Mongolian monetary policy rule, while there has been a limited amount of evidence in the literature. Second, this study examines Mongolian monetary policy rule in a DSGE macroeconomic framework as well as in a single policy reaction function with the GMM estimation, so that the Taylor principle could be identified in a robust manner. Third, this study also estimates the Mongolian policy rate's reaction to exchange rate, so that the degree of the fear of floating could be verified.

The rest of the paper is structured as follows. Section 2 gives an overview of Mongolian monetary policy after the adoption of an inflation targeting in 2007. Section 3 reviews the literature on the studies of monetary policy rules in emerging Asian economies including Mongolia, and highlights this study's contributions. Section 4 conducts the empirical analyses of the Mongolian monetary policy rule by GMM and DSGE estimations with descriptions of the methodology, as well as estimation results and their discussions. Section 5 summarizes and concludes.

## 2. Overview of Mongolian Monetary Policy

This section overviews the trends in Mongolian monetary policy after the adoption of an inflation targeting in 2007. Figure 1 displays the BOM's policy rate and the interest rate corridor, and also compares the actual inflation rate with the targeted rate in terms of annual rate at each year end. The targeted inflation rate was updated by the BOM's Monetary Policy Guidelines for each year.

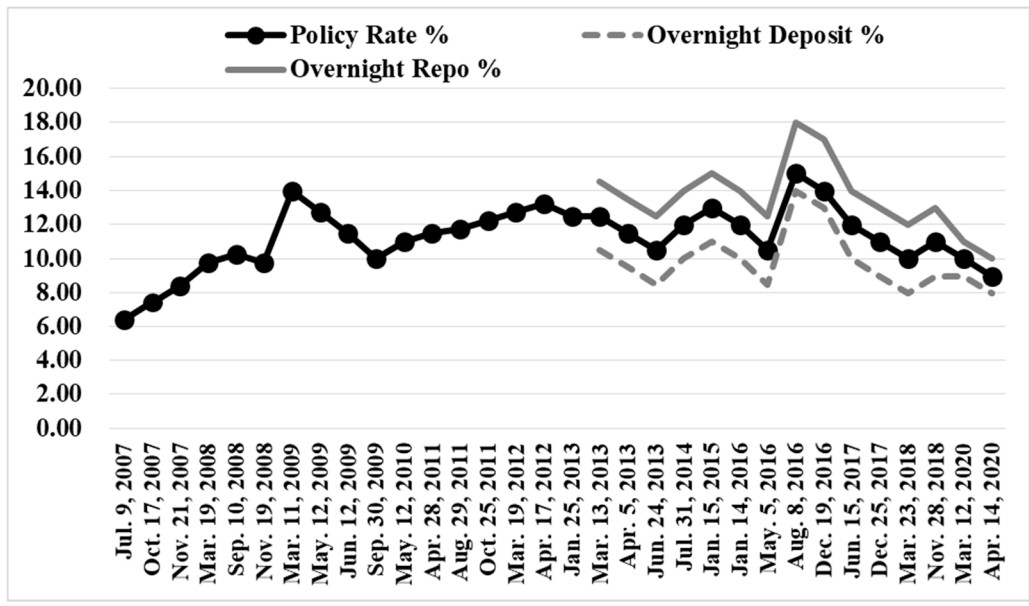

(**a**) Policy Rate and Corridor Rates

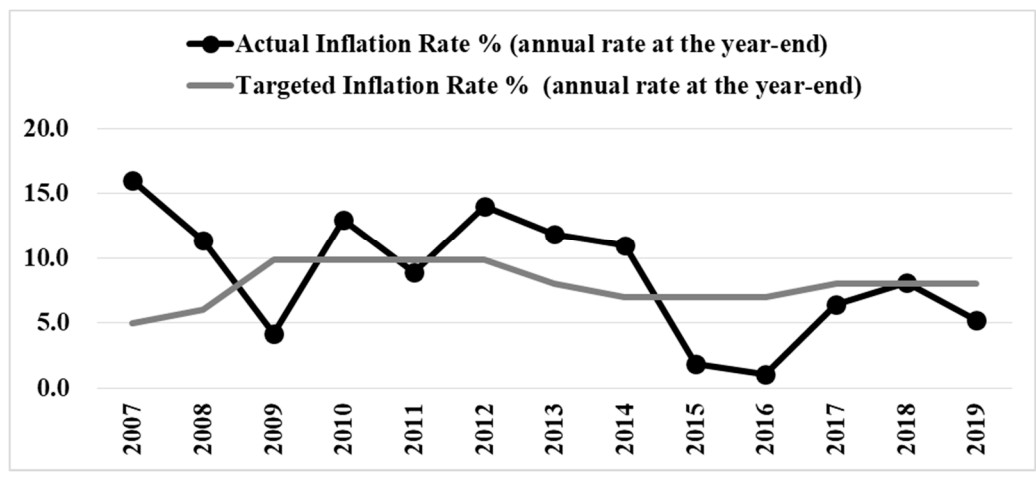

(**b**) Actual and Targeted Inflation Rates

**Figure 1.** Trend in Monetary Policy in Mongolia. Source: Author's description based on the website of the Bank of Mongolia.

Soon after the BOM introduced an inflation targeting in 2007, the Mongolian economy was hit by the world financial crisis in 2009, and the Mongolian government accepted the IMF Stand-by Program in that year. At that time, the BOM raised its policy rate towards 14 percent in March 2009, as there was the need to restore confidence in the local currency and to stop the deposit flight out of its economy. The BOM afterwards reduced its policy rate gradually to 10 percent in September 2009 with the decline in inflation rate.

For the period from 2010 to 2012, the Mongolian economy entered the booming stage with a double-digit inflation rate. The fueling of inflation came from the price elevation of such necessities as food and fuel on the supply side, and the expansionary fiscal policy and the soring of capital inflows in the mining sector on the demand side. Thus the BOM raised its policy rate continuously towards 13.25 percent until January 2013. At the same time, the BOM together with the government initiated the "Medium-term Price Stabilization Program" containing programs to stabilize food and fuel prices in October 2012 to decrease the supply side pressure on inflation.

Since around 2013, however, the Mongolian economy has been getting into a phase of economic slowdown. During 2014–2015, in particular, the net inward foreign direct investment to Mongolia fell significantly (in 2014 by 17 times less than its peak in 2011), due to the downturn of the Chinese economy. Afterwards, the economy with inward foreign direct investment was in a moderate recovery process until 2019. During this phase, the inflation rate calmed down with the rate falling down to one percent level in 2015–2016, and still kept itself below the targeted rate until 2019. The monetary policy in this phase, on the other hand, represented rather complicated reactions: the BOM raised its policy rate in 2014–2015 and in the middle of 2016 to avoid the balance-of-payment crises, while continuous monetary easing was expected under moderate inflation.

As for monetary policy framework, there has been progress in the inflation targeting system. The BOM developed the Forecasting and Policy Analysis System (FPAS) from 2011, aiming at forecast-based policy formation and decision making, and effective communication with the public. The BOM also improved its operational framework by establishing an interest rate corridor from February 2013 as shown in Figure 1.

All in all, the whole period with the inflation targeting since 2007 has two different phases: the first phase had such disturbances as the repercussion from the world financial crisis and the surge of inflation-hike with economic booming; and the second phase experienced economic slow-down and moderate inflation with the improvements in monetary policy frameworks.

## 3. Literature Review and This Study's Contributions

This section reviews the literature on the studies of monetary policy rules in emerging Asian countries including Mongolia, and highlights the contributions of this study. The review organizes the previous studies with a focus on the conformity to the Taylor principle as well as the forward- or backward-looking modes in the policy rules of the emerging Asian countries who have adopted an inflation targeting framework.

The Asian countries who experienced the currency crises in the late 1990s initiated an inflation targeting as their monetary policy frameworks in the post-crisis period: Indonesia in July 2005, Korea in April 1998, the Philippines in January 2002, and Thailand in May 2000. Since they switched their exchange rate regime from a pegged one to a floating one in the crisis times, they intended to utilize an inflation targeting as an alternative anchor for price stability (e.g., Mishkin 2000).

Since then, their monetary policy rules under an inflation targeting have been discussed and examined quantitatively in academic circles. The conformity to the Taylor principle, which denotes that policy rate's response to inflation as being over unity, has been identified in the adopters of an inflation targeting by the following studies: for Indonesia, Hsing (2009), Taguchi and Kato (2011), and Taguchi et al. (2020); for Korea, Kim and Park (2006); for the Philippines, Salas (2006) and Taguchi et al. (2020); and for Thailand, Taguchi and Kato (2011), Lueangwilai (2012), and Taguchi et al. (2020). In terms of the forward- or backward- looking modes in the policy rate's response to inflation, some differences are found among the studies above: for Indonesia, a contemporaneous- and backward-looking rule in Hsing (2009) and Taguchi and Kato (2011) versus a forward-looking rule in Taguchi et al. (2020); and for Thailand, a backward- and contemporaneous- looking rule in Taguchi and Kato (2011) and Lueangwilai (2012) versus a forward-looking rule in Taguchi et al. (2020). These differences might come from the ones in sample periods used in the studies: the upgrading toward forward-looking rules by using updated samples might reflect the recent improvements in forecasting and managing capacities of the adopters of an inflation targeting by accumulating experiences of operating its system. Another category of the development in the studies on monetary policy rules is the identification of a nonlinear Taylor rule. Since Vítor (2011) verified a nonlinear rule in the European Central Bank and Bank of England, the nonlinearity was also identified for Korea by Koo et al. (2012) and for five emerging-market economies including Indonesia, Korea, and Thailand by Caporale et al. (2018). This study, however, does not apply a nonlinear approach due to the lack in sample size in Mongolia.

Regarding the monetary policy rule of Mongolia, one of the adopters of an inflation targeting in Asia, there have been no studies except Taguchi and Khishigjargal (2018), which examined it quantitatively with a policy reaction function. Taguchi and Khishigjargal (2018) described the Mongolian recent rule as an inflation-responsive rule with a forward-looking manner, but with its response weak enough to be pro-cyclical to inflation pressure (against the Taylor principle) due to the "fear of floating".

This study's contribution, particularly related to Taguchi and Khishigjargal (2018), could be highlighted as follows. First, this study re-examines the Mongolian monetary policy reaction function by using updated sample data. Adding the sample period for 2018–2019 on a quarterly basis seems to be critical, since the inflation during that period was well-controlled under the improved management of an inflation targeting as was observed in Section 2. Second, this study has the GMM estimation of a policy reaction function double-checked by the Bayesian estimation of a macroeconomic DSGE model, so that the validity of the Taylor principle could be examined in a robust manner.

## 4. Empirical Analyses of Mongolian Monetary Policy Rule

This section conducts the empirical analyses of the Mongolian monetary policy rule. The section starts with the GMM estimation followed by the DSGE analysis and the discussions of their estimation outcomes.

### 4.1. GMM Estimation

This subsection estimates a monetary policy reaction function by the GMM method for describing the Mongolian monetary policy rule under an inflation targeting with a focus on its conformity to the Taylor principle.

The monetary policy reaction function is specified by the initial work of Clarida et al. (1998) and the subsequent studies such as Belke and Polleit (2007). The original form of the function is denoted by Equation (1), and the empirical specification is presented by Equation (2).

$$r_t^* = \v{r} + \beta \, (E[\pi_{t+n}|\Omega_t] - \pi^*) + \gamma \, (E[y_t|\Omega_t] - y^*) \tag{1}$$

where $r_t^*$ is a target for the central bank's policy rate in period t; $\v{r}$ is a natural rate of nominal interest rate; $\pi_{t+n}$ is an inflation rate between periods t and t + n; $y_t$ is a real output; $\pi^*$ and $y^*$ are respective optimal points for inflation rate and real output; E is an expectation operator; and $\Omega$ is the information available to the central bank at the time it sets a policy rate. Equation (1) is transformed into Equation (2) for an empirical estimation.

$$r_t = (1 - \rho) \, (\alpha + \beta \, \pi_{t+n} + \gamma \, x_t) + \rho \, r_{t-1} + \varepsilon_t \tag{2}$$

where $r_t$, actual policy rate, comes from $r_t = (1 - \rho) \, r_t^* + \rho \, r_{t-1}$ with $\rho \in [0, 1]$ being the degree of policy rate smoothing; the unobserved forecast variables, $E[\pi_{t+n}|\Omega]$ and $E[y_t|\Omega]$, are replaced by the realized variables, $\pi_{t+n}$ and $y_t$; $\alpha$ and $x_t$ are defined as $\alpha \equiv \v{r} - \beta \, \pi^*$ and $x_t \equiv y_t - y^*$ (output gap); and $\varepsilon_t$ is a combination of the central bank's forecast errors of inflation and output, and exogenous disturbances.

Among the parameters, one of the greatest concerns is $\beta$, the degree of policy rate's responsiveness to the inflation rate. In order for the Taylor principle to hold, $\beta$ should be over unity ($\beta > 1$): the policy rate reacts to more than inflation rate, otherwise the real policy rate would accommodate inflation in a pro-cyclical manner. The subscript n of $\pi_{t+n}$ in this study takes the values of 1, 0 and −1 to denote forward-, contemporaneous-, and backward-looking specifications, respectively.

Aside from the policy rate's responses to inflation and output, this study also confirms its reaction to exchange rate to see the degree of the fear of floating.

$$r_t = (1 - \rho) \, (\alpha + \delta \, e_t) + \rho \, r_{t-1} + \varepsilon_t \tag{3}$$

where $e_t$ is a change in exchange rate in terms of local currency (tugriks) value per US dollar. In the case that the central bank prioritizes exchange-rate stabilization in its policy rule, the coefficient δ should take a significantly positive value.

The estimation uses quarterly data running from the third quarter of 2007 (2007Q3) to the present, the fourth quarter of 2019 (2019Q4), during which the BOM operated an inflation targeting. All the data was retrieved from the International Financial Statistics (IFS) of the International Monetary Fund (IMF).[1]

The empirical monetary policy reaction functions in Equations (2) and (3) require the data for the following four indicators: the series of "Central Bank Policy Rate" for policy rate r; "Consumer Prices Index (2010 = 100)" for price index, which is transformed into a year-on-year change rate as inflation rate π; "Industrial Production, Seasonally adjusted, Index (2010 = 100)" for industrial production, which is processed into output gap x by subtracting from the industrial production a Hodrick–Prescott-filter of that series as a proxy of a potential production level; and "National Currency per US Dollar, Period Average" for exchange rate, which is expressed as a year-on-year change rate e.

Before conducting the estimation, the study investigates the stationary property of the data for each variable, by employing the augmented Dickey–Fuller (ADF) unit root test (Said and Dickey 1984) on the null hypothesis that each variable has a unit root in the test equation including "intercept". Table 1 reports the test result for the data for all the indicators, i.e., policy rate r, inflation rate π, output gap x, and exchange rate e for their level data. The test rejected a unit root in all the data at the conventional level of significance by more than 95 percent, thereby their data showed a stationary property. Thus their data are justified to be used for the subsequent estimation.

**Table 1.** Augmented Dickey–Fuller (ADF) Unit Root Test.

| Variable | t-Statistic | Probability |
|:---:|:---:|:---:|
| r | −4.207 *** | 0.001 |
| π | −3.535 ** | 0.011 |
| x | −5.081 *** | 0.000 |
| e | −3.643 *** | 0.008 |

Note: ***, ** denote the rejection of null hypothesis at the 99% and 95% level of significance. Sources: Author's estimation.

For the technique to estimate the parameter vector [α, β, γ, δ, ρ], the study adopts the generalized method of moments (GMM). One of the assumptions required for regression analysis is that the explanatory variables are uncorrelated with the disturbance term. In the case that the equation contains endogenously determined variables as explanatory ones, however, the assumption is violated and the estimator of ordinary least squares is biased and inconsistent. The case could be applied to the estimation Equations (2) and (3) in this study, since the policy interest rate might also affect the explanatory variables. The standard approach to eliminate the effect of variable and residual correlation is to estimate the equation using "instrumental variables" regression. In this context, the GMM estimator is excellent in terms of consistency, asymptotic normality, and efficiency in its property, and has been widely used since the seminal works such as Hansen (1982) and Hansen and Singleton (1982) applied the estimator to their empirical works. Thus this study adopts the GMM estimator and equips the instrumental variables of one-, two- and three-quarter lagged explanatory values of π, x, and e, in the estimation Equations (2) and (3). The J-statistic implies that these instrumental variables are valid in the sense that the over-identifying restrictions cannot be rejected in the models (see Tables 2 and 3).

---

1     The data are retrieved from the website: http://www.imf.org/en/data.

**Table 2.** Estimation Outcomes of Monetary Policy Reaction Functions.

| [Total Period: 2007Q3–2019Q4] | | | |
|---|---|---|---|
| Coefficient | $\pi t - 1$ | $\pi$ | $\pi t + 1$ |
| $(1 - \rho)*\alpha$ | 4.861 (0.693) | −0.657 (−0.053) | 2.753 (0.391) |
| $(1 - \rho)*\beta$ | −0.001 (−0.021) | 0.038 (0.454) | 0.016 (0.255) |
| $(1 - \rho)*\gamma$ | 0.007 (0.455) | 0.013 (0.888) | 0.018 (1.318) |
| $\rho$ | 0.574 (1.009) | 1.031 (1.026) | 0.749 (1.323) |
| J-statistics | 3.766 (0.287) | 0.707 (0.871) | 0.316 (0.956) |
| Long-term Coefficients | | | |
| $\alpha$ | 11.414 | - | 11.001 |
| $\beta$ | -0.003 | - | 0.064 |
| $\gamma$ | 0.018 | - | 0.074 |
| [First-half Period: 2007Q3–2011Q4] | | | |
| Coefficient | $\pi t$-1 | $\pi$ | $\pi t + 1$ |
| $(1 - \rho)*\alpha$ | 9.415 *** (9.284) | 0.804 (0.222) | 10.110 ** (2.565) |
| $(1 - \rho)*\beta$ | −0.066 *** (-5.538) | 0.016 (0.568) | −0.048 (−1.169) |
| $(1 - \rho)*\gamma$ | −0.008 * (−2.020) | 0.001 (0.123) | −0.005 (−0.522) |
| $\rho$ | 0.226 ** (2.670) | 0.940 ** (2.955) | 0.136 (0.419) |
| J-statistics | 3.760 (0.288) | 1.238 (0.743) | 1.924 (0.588) |
| Long-term Coefficients | | | |
| $\alpha$ | 12.174 *** | 13.429 | 11.706 ** |
| $\beta$ | −0.086 *** | 0.278 | −0.056 |
| $\gamma$ | −0.011 * | 0.017 | −0.006 |
| [Second-half Period: 2012Q1–2019Q4] | | | |
| Coefficient | $\pi t - 1$ | $\pi$ | $\pi t + 1$ |
| $(1 - \rho)*\alpha$ | 2.461 (0.430) | 0.183 (0.074) | 0.125 (0.029) |
| $(1 - \rho)*\beta$ | 0.083 ** (2.362) | 0.110 *** (2.927) | 0.067 ** (2.351) |
| $(1 - \rho)*\gamma$ | 0.086 * (1.717) | 0.139 (1.066) | 0.002 (0.069) |
| $\rho$ | 0.736 (1.542) | 0.905 *** (4.007) | 0.923 ** (2.606) |
| J-statistics | 3.040 (0.385) | 1.034 (0.793) | 3.631 (0.2C9) |

**Table 2.** *Cont.*

| Long-term Coefficients | | | |
| --- | --- | --- | --- |
| α | 9.334 | 1.954 | 1.653 |
| β | 0.316 ** | 1.172 *** | 0.885 ** |
| γ | 0.327 * | 1.480 | 0.031 |

Note: ***, **, * denote the rejection of null hypothesis at the 99%, 95%, and 90% level of significance. The numbers in parentheses are *t*-values, except that those in J-statistics are their probabilities. Sources: Author's estimation.

**Table 3.** Monetary Policy Reaction to Exchange Rate.

| Coefficient | 2007q3–2019q4 | 2007q3–2011q4 | 2012q1–2019q4 |
| --- | --- | --- | --- |
| $(1 - \rho)^*\alpha$ | −0.231 (−0.036) | 2.652 ** (2.905) | 9.574 (0.487) |
| $(1 - \rho)^*\delta$ | −0.039 (−0.939) | −0.051 * (−1.914) | 0.022 (0.448) |
| ρ | 1.053 * (1.789) | 0.790 *** (6.874) | 0.167 (0.100) |
| J-statistics | 0.158 (0.690) | 0.131 (0.716) | 0.588 (0.442) |
| Long-term Coefficients | | | |
| α | - | 12.685 ** | 11.495 |
| δ | - | −0.244 * | 0.027 |

Note: ***, **, * denote the rejection of null hypothesis at the 99%, 95%, and 90% level of significance. The numbers in parentheses are t-values, except that those in J-statistics are their probabilities. Sources: Author's estimation.

The GMM estimation is conducted for the total sample period (2007Q3–2019Q4), and also for the first half (2007Q3–2011Q4) and the second half (2012Q1–2019Q4) periods, since the whole period with an inflation targeting has two different phases as was described in Section 2: the first phase with economic disturbances and high inflation, and the second phase with moderate inflation and policy improvements. The breakpoint in the total sample is set at 2012Q1 following the previous study of Taguchi and Khishigjargal (2018), and this study also reconfirmed the breakpoint statistically by the Chow's breakpoint test: the F-statistic (7.166) rejected the hypothesis of parameter stability over different periods with the breakpoint being 2012Q1 with a probability of more than 99 percent.

Table 2 reports the estimation outcomes of monetary policy reaction functions with forward-, contemporaneous-, and backward- looking specifications for different sample periods: the total, the first part and the second part ones. Table 3 shows the reaction to the change in exchange rate for three different sample periods. In each table, based on the estimated short-term coefficients in the upper part, the long-term coefficients [α, β, γ] are computed and displayed in the lower part.

Focusing on the long-term coefficients in Table 2, it is only in the second-half-period estimation that the coefficients of inflation are positive at the significant level of more than 95 percent, and more importantly, the coefficient is beyond unity in the contemporaneous-looking specification (β = 1.172 in $\pi_t$), implying the conformity to the Taylor principle. The other coefficients including those of output gap are insignificant or weakly significant, otherwise impossible to calculate since the degree of smoothing ρ is over unity. As for Table 3, no meaningful coefficients are found in the estimation on the policy rate' responses to exchange rate fluctuations.

All in all, the Mongolian monetary policy rule in the recent phase of an inflation targeting is characterized by an inflation-responsive rule fulfilling the Taylor principle, and the fear of floating is not serious enough to disturb the inflation-responsive rule.

*4.2. New Keynesian DSGE Estimation*

This section turns to a New Keynesian DSGE estimation in order to re-check the validity of the Taylor principle verified by the GMM estimation in the previous section. This section first specifies the model structure, and then presents the estimation result.

A New Keynesian DSGE model, which was developed by Gali (2008), was built on the micro-founded characteristic of a Real Business Cycle model (Kydland and Prescott 1982) with nominal rigidities. In the virtue of the advances of the estimation technique, especially since the seminal works in Christiano et al. (2005) and Smets and Wouters (2003, 2007), a New Keynesian model has been widely used for macroeconomic studies during the recent decades. One of the extensions of the simple New Keynesian model is to model a small open economy aside from a closed economy. Gali and Monacelli (2005), for instance, laid out a small open economy version of a model with Calvo-type staggered price-setting and with the equilibrium conditions reflecting degree of openness and world output fluctuations, and used it for analyzing macroeconomic implications of alternative monetary policy regimes including an exchange rate peg.

This study applies the small open economy version of a New Keynesian DSGE model to examine the Mongolian monetary policy rule, since Mongolian economy is considered to be a typical small open economy.[2] The estimable model consists of the following ten equations based on Gali and Monacelli (2005).

$$\widetilde{x}_t = E_t[\widetilde{x}_{t+1}] - (1/\sigma_\alpha)\left(\widetilde{r}_t - E_t[\pi_{H,t+1}] - \overline{rr}\, n_t\right) \tag{4}$$

$$\overline{rr}_t = -\sigma_\alpha \Gamma\left(1 - \rho_a\right)a_t + \alpha\, \sigma_\alpha (\Theta + \Psi)\, E_t[\Delta \widetilde{y}^*_{t+1}] \tag{5}$$

$$\pi_{H,t} = \beta\, E_t[\pi_{H,t+1}] + \kappa_\alpha \widetilde{x}_t + e_t \tag{6}$$

$$\widetilde{r}_t = \phi_r \widetilde{r}_{t-1} + (1 - \phi_r)\left(\phi_\pi \pi_t + \phi_x \widetilde{x}_t\right) + \varepsilon_{rt} \tag{7}$$

$$\pi_t = \pi_{H,t} + \alpha\, \Delta s_t \tag{8}$$

$$s_t = \sigma_\alpha\left(\widetilde{y}_t - \widetilde{y}^*_t\right) \tag{9}$$

$$\widetilde{y}_t = \widetilde{x}_t + \left(\Gamma a_t + \alpha\, \Psi \widetilde{y}^*_t\right) \tag{10}$$

$$a_t = \rho_a\, a_{t-1} + \varepsilon_{at} \tag{11}$$

$$e_t = \rho_e\, e_{t-1} + \varepsilon_{et} \tag{12}$$

$$\widetilde{y}^*_t = \rho_w \widetilde{y}^*_{t-1} + \varepsilon_{wt} \tag{13}$$

The list of endogenous and exogenous variables, and the one of fixed and estimated parameters including definition identities, are presented in Tables 4 and 5, respectively. The variables in the log-linearized version are expressed by the percentage deviation from the zero-inflation steady-state level.

---

[2] The Mongolian economic size is 13.0 billion US dollars at current prices in terms of GDP in 2018, while the average size of Asian developing economies is 621.7 billion US dollars in the same year. The Mongolian ratio of "imports of goods and services" to GDP is 55.6 percent on the average during the period from 1995 to 2018, while the average ratio in Asian developing economies is 35.1 percent during the same period. The data of the GDP and the ratio of "imports of goods and services" to GDP are retrieved respectively from UNCTAD STAT: http://unctadstat.unctad.org/EN/.

**Table 4.** List of Endogenous and Exogenous Variables.

| | [Endogenous Variables] |
|---|---|
| $\widetilde{x}$ | Output gap |
| $\widetilde{y}$ | Output |
| $\pi$ | CPI inflation (the rate of change in consumer prices) |
| $\pi_H$ | Domestic inflation (the rate of change in domestic goods prices) |
| $\widetilde{r}$ | Nominal interest rate |
| $\overline{\overline{rr}}$ | Natural rate of interest rate |
| s | Terms of trade |
| E | Expectation operator |
| | [Exogenous Variables] |
| $\widetilde{y}^*$ | World output that follows first-order autoregressive with i.i.d. shock, $\varepsilon_w$ |
| a | Productivity shock that follows first-order autoregressive with i.i.d. shock, $\varepsilon_a$ |
| e | Cost-push shock that follows first-order autoregressive with i.i.d. shock, $\varepsilon_e$ |
| $\varepsilon_r$ | Monetary policy shock with i.i.d. |

Descriptions (~denotes the deviation from the steady-state level)

**Table 5.** List of Parameters.

| [Fixed Parameters] | Descriptions | Assumption | Notes |
|---|---|---|---|
| $\alpha$ | Degree of economic openness | 0.58 | Import/GDP ratio in the sample average |
| $\beta$ | Discount factor for households | 0.99 | |
| $\gamma$ | Substitutability between goods produced in different foreign countries | 1.00 | |
| $\eta$ | Substitutability between domestic and foreign goods | 1.00 | |
| $\theta$ | Probability a firm does not change its price | 0.75 | |
| $\sigma$ | Parameter on utility of consumption under constant relative risk aversion (CRRA) | 1.00 | Log utility of consumption |
| $\varphi$ | Parameter on disutility of labor | 0.00 | Linear disutility of labor |
| $\rho_a$ | Autoregressive parameter for productivity shock | 0.90 | |
| $\rho_e$ | Autoregressive parameter for cost-push shock | 0.90 | |
| $\rho_w$ | Autoregressive parameter for world GDP shock | 0.90 | |
| **[Definitional Identities]** | | | |
| $\kappa_\alpha \equiv \lambda\,(\sigma_\alpha + \varphi)$ | | | |
| $\lambda \equiv \{(1 - \beta\,\theta)\,(1 - \theta)\}/\theta$ | | | |
| $\sigma_\alpha \equiv \sigma/(1 - \alpha) + \alpha\,\omega$ | | | |
| $\omega \equiv \sigma\,\gamma + (1 - \alpha)\,(\sigma\,\eta - 1)$ | | | |
| $\Gamma \equiv (1 + \varphi)/(\sigma_\alpha + \varphi)$ | | | |
| $\Theta \equiv (\sigma\,\gamma - 1) + (1 - \alpha)\,(\sigma\,\eta - 1)$ | | | |
| $\Psi \equiv -\,\Theta\,\sigma_\alpha/(\sigma_\alpha + \varphi)$ | | | |
| **[Estimated Parameters: Monetary policy rule]** | | | |
| $\phi_r$ | Smoothing degree of policy rate | | |
| $\phi_\pi$ | Policy rate reaction to CPI inflation | | |
| $\phi_x$ | Policy rate reaction to output gap | | |

The first four equations from (4) to (7) constitute the major structure of a New Keynesian model (with a small open economy version), characterizing the dynamic behavior of three key macroeconomic indicators: output gap, inflation, and nominal interest rate. Equation (4), called the "expectational IS curve", corresponds to the log-linearization of an optimizing household's Euler equation. Equation (5)

represents the determination of the natural rate of interest rate. Equation (6), called the New-Keynesian Phillips curve, describes the optimizing behavior of monopolistically competitive firms that set their prices in a randomly staggered fashion, as suggested by Calvo (1983). Equation (7) represents the monetary policy rule, corresponding to Equations (1) and (2) shown in Section 4.1. The subsequent three equations from (8) to (10) describe the nexus between CPI inflation (the change in consumer prices) and domestic inflation (the change in domestic goods prices), representing the property of a small open economy, i.e., the linkage between a small open economy and the world economy through economic openness and terms of trade.[3]

This paper uses the Bayesian method to estimate the parameters of the model above, though some parameters are fixed in advance.[4] For more details of the estimation, see, for example, An and Schorfheide (2007). Regarding the observed data, this DSGE estimation uses them for the three endogenous variables: output gap $\widetilde{x}$, domestic inflation $\pi_H$, and nominal interest rate $\widetilde{r}$. The data for the domestic inflation (the change in domestic goods prices) are calculated by a year-on-year change in the GDP deflator obtained by the division between nominal GDP and GDP at constant prices (retrieved from the National Statistics Office of Mongolia[5]). As for the data for output gap and nominal interest rate, the data of output gap x and policy rate r in Section 4.1 are applied, although the data of policy rate is processed into a detrended series by subtracting a Hodrick–Prescott-filter of that data, since the model is expressed by the deviation from the steady-state level.[6] The sample period corresponds to the second-half one in Section 4.1.

This study focuses on the estimation of the parameters that appear in the monetary policy rule in Equation (7), namely, $[\phi_r, \phi_\pi, \phi_x]$, and thus the other parameters are treated as fixed. As shown in Table 5, the parameter on the degree of economic openness $\alpha$ is set to 0.58, which corresponds to the import/GDP ratio on the average in the sample period[7]. Second, the parameters $[\beta, \gamma, \eta, \theta, \sigma, \varphi, \rho_a, \rho_e, \rho_w]$ are set according to various types of DSGE literature studies such as Smets and Wouters (2003, 2007), Gali and Monacelli (2005), Gali (2008). Finally, the parameters $[\kappa_\alpha, \lambda, \sigma_\alpha, \omega, \Gamma, \Theta, \Psi]$ are set in the same way as Gali and Monacelli (2005). The prior means of $[\phi_r, \phi_\pi, \phi_x]$ are set to the values estimated by the GMM in Section 4.1. Table 6 reports the prior-value settings in the left side of the column. The prior means of the parameters on the reaction to inflation $\phi_\pi$ and the smoothing degree $\phi_r$ correspond to the GMM-estimated parameters of the case $\pi$ in the second half sample period ($\beta = 1.172$ and $\rho = 0.905$), which satisfy the Taylor principle. The prior-mean-value of the parameter on the reaction to output gap $\phi_x$ is set to zero, however, as the GMM-estimated coefficient was insignificant in that case.

The outcomes of the Bayesian estimations are summarized in terms of the posterior distributions in Table 6 and Figure 2. In the comparison between prior and posterior distributions in Table 6, the shift away from the priors to the posteriors implies that the observed data add important information to the estimation of the posteriors. It is worth noting that the posterior means of parameters on the reaction to inflation $\phi_\pi$ and the smoothing degree $\phi_r$ have almost the same values as their prior means: in the reaction to inflation, 1.152 (posterior) versus 1.172 (prior); and in the smoothing degree, 0.891 (posterior) versus 0.905 (prior). Regarding the reaction to output gap, the posterior means turns out to be positive but still insignificant, judging from the Highest Posterior Density (HPD) Interval with its bottom line being negative.

---

[3] The equations from (4) to (10) except (7) correspond to those in Gali and Monacelli (2005) as follows: Equations (4) and (5) to (37) in p. 719 of Gali and Monacelli (2005); (6) to (36) in p. 718; (8) to (14) in p. 712; (9) to (29) in p. 717; and (10) to (35) in p. 718.

[4] For the Bayesian estimation, the study uses the software of Dynare and Matlab.

[5] See the Mongolian Statistical Information Service at the website: https://www.en.nso.mn/.

[6] The data of domestic inflation and output gap have no need to be processed under the assumption of zero-inflation steady-state.

[7] The import/GDP ratio is the one that divides "Imports of Goods and Services" by GDP in Mongolia, using the data of International Financial Statistics of International Monetary Fund.

**Table 6.** Dynamic Stochastic General Equilibrium (DSGE) Bayesian Estimation.

| Parameters | | Priors | | | Posterior | |
|---|---|---|---|---|---|---|
| | | Dist. | Mean | Stdev. | Mean | 90% HPD Interval |
| Monetary policy rule | | – | – | – | – | – |
| Inflation | $\phi_\pi$ | norm | 1.172 | 0.050 | 1.152 | 1.071–1.232 |
| GDP gap | $\phi_x$ | norm | 0.000 | 0.050 | 0.011 | −0.003–0.027 |
| Smoothing | $\phi_r$ | norm | 0.905 | 0.050 | 0.891 | 0.854–0.928 |
| Shocks | | | | | | |
| Monetary Policy | $\varepsilon_{rt}$ | invg | 1.000 | 1.000 | 1.276 | 0.764–1.777 |
| Productivity | $\varepsilon_{at}$ | invg | 1.000 | 1.000 | 0.927 | 0.313–1.607 |
| Cost-push | $\varepsilon_{et}$ | invg | 1.000 | 1.000 | 2.124 | 1.658–2.567 |
| World GDP | $\varepsilon_{wt}$ | invg | 1.000 | 1.000 | 17.176 | 13.101–22.539 |

Sources: Author's estimation.

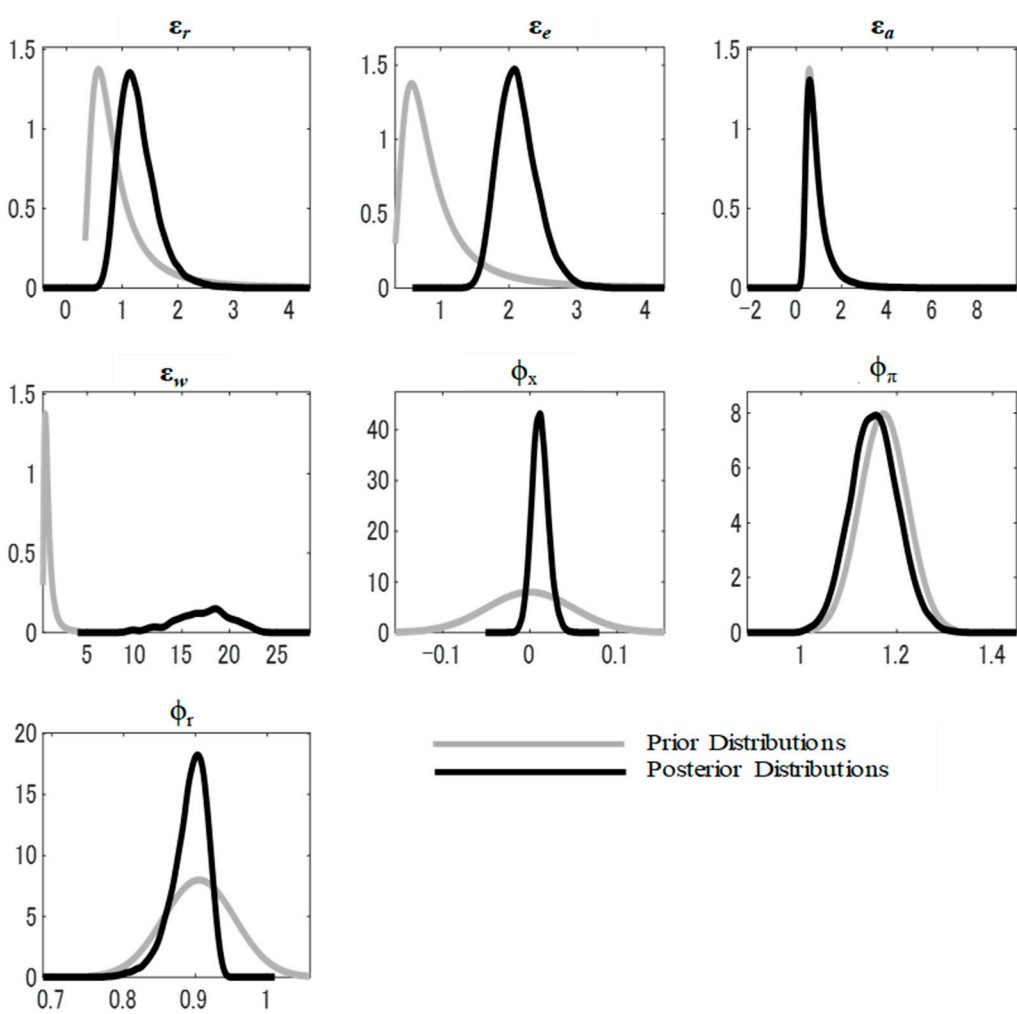

**Figure 2.** Prior and Posterior Distributions. Source: Author's estimation.

Figure 3 shows the impulse response functions to monetary policy shock (one percent point of nominal interest rate shock). It shows that CPI inflation as well as domestic inflation respond negatively to monetary policy shock over ten quarters. It should be noted that the negative response of CPI

inflation is sharper than that of domestic inflation, since the negative impact of terms of trade is added on to the CPI inflation response as Equation (8) in the model suggests.

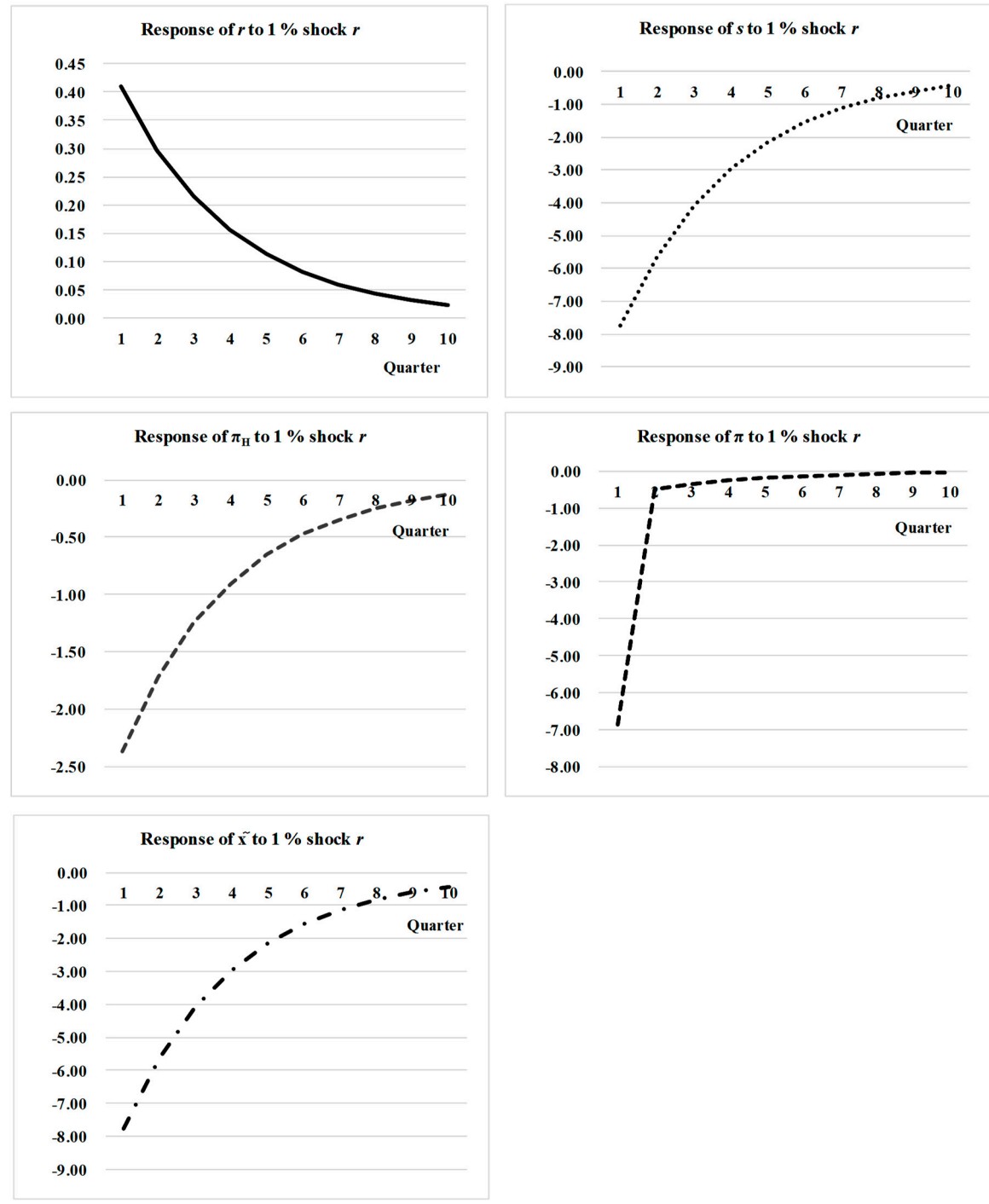

**Figure 3.** Impulse Responses to Monetary Policy Shock under the Dynamic Stochastic General Equilibrium (DSGE) Model. Source: Author's estimation.

In sum, the Bayesian estimation of the New Keynesian DSGE model with a small open economy version could endorse the GMM estimation of policy reaction function on the monetary policy rule in Mongolia, in the sense that the outcome of both the estimations on the policy rate reaction to inflation are similar.

*4.3. Discussions on Estimation Outcomes*

This section discusses how to interpret the estimation outcomes in the context of the Mongolian official monetary policy stance and in comparison with the previous studies presented in Section 3. Both of the GMM and DSGE estimations in Sections 4.1 and 4.2 identified an inflation-responsive monetary policy rule fulfilling the Taylor principle in Mongolia. This result is consistent with the Mongolian policy mandate of an inflation targeting in that the BOM should take all possible measures to attain the targeted inflation through the policy rate operation. In particular, the conformity to the Taylor principle confirmed for the second half sample period fits well with the upgraded inflation targeting called the FPAS that has been adopted since 2011.

Compared with the previous studies on the Mongolian monetary policy rule, this study and Taguchi and Khishigjargal (2018) commonly verifies an inflation-responsive rule, but it is in this study, not in Taguchi and Khishigjargal (2018), that the conformity to the Taylor principle is identified. This is probably due to this study's updating of the sample data by adding the period for 2018–2019 with the inflation being well-controlled under the improved management of an inflation targeting. The strength of the policy rate reaction to inflation in Mongolia could also be compared with those in the other emerging Asian countries and an advanced country like the US. The Mongolian coefficient of inflation responsiveness is estimated to be under 1.2 in both GMM and DSGE approaches in this study. It is rather a weaker reaction compared with those of the other Asian adopters of an inflation targeting according to the latest study of Taguchi et al. (2020): 1.3 in Thailand, 1.4 in the Philippines and 1.8 in Indonesia, and further with the US Fed reaction, 2.27–2.57 exhibited by Belke and Polleit (2007). Thus there seems to be still room to investigate whether the Mongolian policy rate reaction to inflation, although fulfilling the Taylor principle, would be powerful enough to control inflation in the case of its high pressure.

This study focused on Mongolia to examine a monetary policy rule through the GMM and DSGE approaches. These approaches could also be applied to the investigation of monetary policy rules in the other emerging market economies, as they have improved their inflation targeting management. In fact, Taguchi et al. (2020) adopted the GMM and DSGE approaches for analyzing the monetary policy rules in Indonesia, the Philippines and Thailand, although their New Keynesian DSGE estimation was based on a closed economy's version. It is expected that these approaches will be used widely for examining monetary policy rules in emerging-market economies with extended versions of a New Keynesian DSGE model.

## 5. Concluding Remarks

This article examined the monetary policy rule under an inflation targeting in Mongolia with a focus on its conformity to the Taylor principle, through two kinds of approaches: a monetary policy reaction function by the GMM estimation and a New Keynesian DSGE model with a small open economy version by the Bayesian estimation. This study contributes to the enrichment of evidence in assessing an inflation targeting adopted by emerging market economies. The main findings are summarized as follows. First, the GMM estimation identified the contemporaneous inflation-responsive rule fulfilling the Taylor principle in the recent phase of an inflation targeting. Second, the DSGE-model estimation endorsed the GMM estimation by producing a consistent outcome on the monetary policy rule. Third, the Mongolian rule was estimated to have a weaker response to inflation than the rules of the other emerging Asian adopters of an inflation targeting.

**Author Contributions:** Conceptualization, methodology and formal analysis, H.T. and G.G.; investigation and data curation, G.G.; writing—original draft preparation and writing—review and editing, H.T.; All authors have read and agreed to the published version of the manuscript.

**Funding:** This research received no external funding.

**Acknowledgments:** We appreciate Kenichi Tamegawa, Yamagata University, for his contribution to inputting the necessary knowledge in the estimation of the DSGE model with the small open economy version.

**Conflicts of Interest:** The authors declare no conflict of interest.

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
