# Peer review of "Monetary Policy Rule and Taylor Principle in Mongolia: GMM and DSGE Approaches"

_ijfs, doi:10.3390/ijfs8040071_

Round 1

Reviewer 1 Report

In my opinion, the article meets all the requirements for scientific articles.

The article defines the goal, theoretical and empirical literature is reviewed, appropriate econometric tools were used, with the use of an appropriate model verification method. The article takes into account the conclusions of the research and refers to the polemics and opinions on the topic under study.

In addition, the article appears to be linguistically correct, is written specifically and adds value to Mongolian monetary policy on inflation issues.

I support publishing a peer-reviewed article.

Author Response

In my opinion, the article meets all the requirements for scientific articles.

The article defines the goal, theoretical and empirical literature is reviewed, appropriate econometric tools were used, with the use of an appropriate model verification method. The article takes into account the conclusions of the research and refers to the polemics and opinions on the topic under study.

In addition, the article appears to be linguistically correct, is written specifically and adds value to Mongolian monetary policy on inflation issues.

I support publishing a peer-reviewed article.

--> We appreciated the high evaluation for our manuscript. We have given our manuscript thorough proofreadings sentence by sentence.  

Reviewer 2 Report

This paper applies GMM and DSGE Approaches to study the examine the monetary policy rule under inflation targeting in Mongolia. They have done a complete study and have several interesting findings. I think the overall quality of this paper is good and I suggest the publication after a minor revision. I list my comments below.

First, I suggest the author to do another round proof reading and fix the minor grammar mistakes and issues. 

Second, the figure and table provided in this paper is low quality and I suggest the author to revise for better display. 

Third, for the GMM estimation rule, I think the authors should first give a brief background. Then the effects of the parameters on the model performance should be studied and I don't see it in the current version. I suggest the authors to deep dive regarding this point.

Fourth, they focus on the Mongolia itself, and I am wondering if GMM and DSGE Approaches can be applied to other similar countries and suggest the author to discuss about it.

Author Response

This paper applies GMM and DSGE Approaches to study the examine the monetary policy rule under inflation targeting in Mongolia. They have done a complete study and have several interesting findings. I think the overall quality of this paper is good and I suggest the publication after a minor revision. I list my comments below.

First, I suggest the author to do another round proof reading and fix the minor grammar mistakes and issues.

-> We have done thorough proofreadings and corrected several typos. Please see the track-changes in the revised manuscript. 

 Second, the figure and table provided in this paper is low quality and I suggest the author to revise for better display. 

-> We have enlarged the figures and tables for better display.Please see Table 2-6 and Figure 2-3.

Third, for the GMM estimation rule, I think the authors should first give a brief background. Then the effects of the parameters on the model performance should be studied and I don't see it in the current version. I suggest the authors to deep dive regarding this point.

-> We have added the description to explain the background that would justify the usage of the GMM estimation. Please see the line 235-246 in red-ink in the track-changes version of the revised manuscript. 

Fourth, they focus on the Mongolia itself, and I am wondering if GMM and DSGE Approaches can be applied to other similar countries and suggest the author to discuss about it.

-> We have added the discussion on the application of GMM and DSGE approaches to the other emerging-market economies. Please see the line 417-424 in red ink in the track-changes version of the revised manuscript. 

Reviewer 3 Report

I would suggest to expand the literature review part in order to analyze different views. Also, I would suggest to include more references in the reference list and to pay more attention to the newest studies. 

Author Response

I would suggest to expand the literature review part in order to analyze different views. Also, I would suggest to include more references in the reference list and to pay more attention to the newest studies.

-> We have expanded the literature by adding the studies on a nonlinear Taylor rule: Vitor (2011), Koo et al. (2012) and Caporale et al. (2018), and included them in the reference list. Please see the line 160-165 in red ink in the track-changes version of the revised manuscript, and also check the reference list in Page 15 in red ink.